# Point-of-care infrared thermal imaging for differentiating venomous snakebites from non-venomous and dry bites

**Paramasivam Sabitha, Chanaveerappa Bammigatti, Surendran Deepanjali, Bettadpura Shamanna Suryanarayana, Tamilarasu Kadhiravan** *

Department of Medicine, Jawaharlal Institute of Postgraduate Medical Education and Research, Puducherry, India

* kadhir@jipmer.edu.in

## Abstract

### Background

Local envenomation following snakebites is accompanied by thermal changes, which could be visualized using infrared imaging. We explored whether infrared thermal imaging could be used to differentiate venomous snakebites from non-venomous and dry bites.

### Methods

We prospectively enrolled adult patients with a history of snakebite in the past 24 hours presenting to the emergency of a teaching hospital in southern India. A standardized clinical evaluation for symptoms and signs of envenomation including 20-minute whole-blood clotting test and prothrombin time was performed to assess envenomation status. Infrared thermal imaging was done at enrolment, 6 hours, and 24 hours later using a smartphone-based device under ambient conditions. Processed infrared thermal images were independently interpreted twice by a reference rater and once by three novice raters.

### Findings

We studied 89 patients; 60 (67%) of them were male. Median (IQR) time from bite to enrolment was 11 (6.5–15) hours; 21 (24%) patients were enrolled within 6 hours of snakebite. In all, 48 patients had local envenomation with/without systemic envenomation, and 35 patients were classified as non-venomous/dry bites. Envenomation status was unclear in six patients. At enrolment, area of increased temperature around the bite site (Hot spot) was evident on infrared thermal imaging in 45 of the 48 patients with envenomation, while hot spot was evident in only 6 of the 35 patients without envenomation. Presence of hot spot on baseline infrared thermal images had a sensitivity of 93.7% (95% CI 82.8% to 98.7%) and a specificity of 82.9% (66.3% to 93.4%) to differentiate envenomed patients from those without envenomation. Interrater agreement for identifying hot spots was more than substantial (Kappa statistic >0.85), and intrarater agreement was almost perfect (Kappa = 0.93). Paradoxical thermal changes were observed in 14 patients.

**Data Availability Statement:** All relevant data are within the manuscript and its Supporting Information files.

**Funding:** The authors received no specific funding for this work.

**Competing interests:** The authors have declared that no competing interests exist.

## Conclusions

Point-of-care infrared thermal imaging could be useful in the early identification of non-venomous and dry snakebites.

### Author summary

Most venomous snakebites cause swelling of the bitten body part within a few hours if venom had been injected. Usually, health care providers diagnose venomous snakebites by doing a clinical examination and by testing for incoagulable blood. If no abnormalities are found, then the snakebite is diagnosed as a non-venomous bite or a dry bite. Swelling of the bitten body part results from venom-induced inflammation and is accompanied by local increase in skin temperature. It is possible to capture visual images of these temperature changes by using infrared imaging, the same technology used in night vision cameras. This study found that most persons with venomous snakebites had hot areas on infrared images while such changes were observed in only a few persons with non-venomous or dry snakebites. This new knowledge could help doctors identify non-venomous and dry snakebites early.

## Introduction

About 5 million people are bitten by venomous and non-venomous snakes every year, and it has been estimated that snakebite envenomation results in about 81,000 to 138,000 deaths globally [1]. The assessment whether a snakebite victim is envenomed or not, particularly in settings with limited laboratory capacity, is essentially based on clinical evaluation for symptoms and signs of local and systemic envenomation and a 20-minute whole-blood clotting test (WBCT20) [2]. Bites by vipers and some cobras result in local swelling within a few hours, and swelling of the bitten limb is a definitive sign of local envenomation [2]. However, clinical assessment of local swelling and warmth is often equivocal, especially when seen early on after a snakebite. Likewise, despite its simplicity, the WBCT20 suffers from frequent false-positive results leading to inappropriate use of precious antivenom [3,4]. Thus, there is an unmet need for a simple, rapid, and objective assessment tool that could differentiate venomous snakebites from non-venomous and dry bites.

While exploring the use of infrared thermal imaging to assess progression of local swelling caused by snakebites, we came across an interesting report by Medeiros et al. in which changes were evident on infrared thermal imaging in two patients with snakebite envenomation, whereas one patient with dry bite inflicted by a lance-headed pit viper did not exhibit thermal changes [5]. Based on this limited observation, Medeiros et al. had suggested using infrared thermal imaging to identify dry bites. We, therefore, conducted the present study to evaluate whether infrared thermal imaging could be used to differentiate venomous snakebites from non-venomous and dry bites.

## Methods

### Ethics statement

The study protocol was reviewed and approved by the Institute Ethics Committee (Human studies) at Jawaharlal Institute of Postgraduate Medical Education and Research (JIPMER),

Puducherry (Approval No. JIP/IEC/2018/0384; 11/04/2018). We obtained informed written consent from all study participants.

## Setting and population

This study was conducted during the period October 2018 through October 2019 at JIPMER, Puducherry located in southern India. Catchment area of JIPMER hospital includes the union territory of Puducherry and the four adjoining districts in the state of Tamil Nadu (Fig 1). Geography of the catchment area is largely made of plain terrain and coastal plains with a few hillocks at some places, and has a tropical savanna climate with dry winter. Of the 23 venomous terrestrial snake species documented in India, 8 are found in Tamil Nadu and Puducherry [6]. Of the eight, four species (*Calliophis beddomei*, *Calliophis nigrescens*, *Hypnale hypnale*, and *Ophiophagus hannah*) are restricted to the Western and Eastern Ghats (mountain ranges), far away from the study catchment area [6]. Venomous snake species documented in the study catchment area are: Russell's viper (*Daboia russelii*), saw-scaled viper (*Echis carinatus*), spectacled cobra (*Naja naja*), and common krait (*Bungarus caeruleus*). In addition, a few medically important non-front-fanged colubroid (NFFC) snake species—common vine snake (*Ahaetulla nasuta*), common cat snake (*Boiga trigonata*), and Forsten's cat snake (*Boiga forsteniare*)—are found in the study area [6].

We enrolled patients aged 18 years or more presenting to the Emergency Medical Services with a definitive history of snakebite in the past 24 hours. A history of snake bite was considered definitive if the victim was sure that the offending animal was a snake (may or may not be able to identify the species) and that it had actually bitten the victim. We excluded patients if

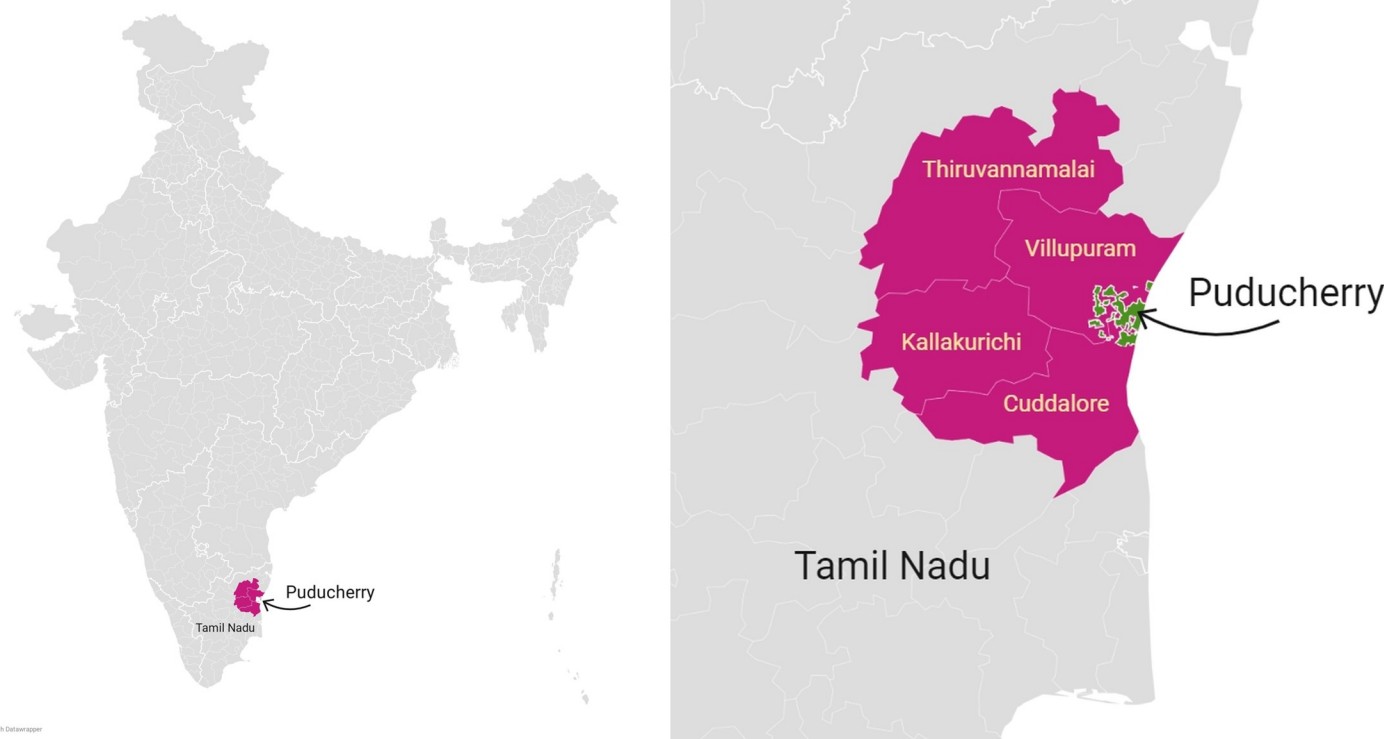

**Fig 1. Study catchment area highlighted on political map of India.** It includes union territory of Puducherry, where the study hospital is located, and four adjoining districts in the state of Tamil Nadu. Courtesy: https://www.datawrapper.de/.

they had received antivenom treatment before study enrolment, gave a history of wound manipulation, or were seriously ill.

After obtaining consent, one of the investigators (PS) evaluated the patients for symptoms and signs of local and systemic envenomation using a standardized data collection form. Physical findings were independently verified later by another investigator (TK). Bite site was carefully examined for fang marks, bleeding, and features of inflammation such as pain, swelling, erythema, warmth, necrosis, and blebs. Presence of regional lymphadenitis was also noted. One of the investigators (PS) performed a WBCT20 using a fresh clean borosilicate glass tube as described earlier [3]. Prothrombin time was tested in the clinical laboratory on blood samples collected into citrate tubes. An international normalized ratio (INR) >1.40 was considered abnormal [3]. Decision to administer antivenom was made by the treating units in accordance with their usual practice. If there was no evidence of envenomation at presentation, patients were re-evaluated 6 hours and 24 hours later. Patients without evidence of envenomation up to 24 hours were discharged home. Finally, we adjudicated the envenomation status using an interpretation guide (Table 1). We developed this interpretation guide based on the World Health Organization guidelines for the management of snakebites in south-east Asia [2] and earlier studies on diagnostic accuracy of WBCT20 [3,4], to ensure uniformity in adjudication of envenomation status for study purpose.

We identified the offending snake species by examining the specimen or photograph of dead snake, when brought by patients. Otherwise, patients were asked to identify the snake on a template picture. If the snake could not be identified by these methods, we recorded as 'unidentified' snake species.

## Infrared thermal imaging and analysis

Infrared thermal imaging was done by an investigator (PS) at enrolment, 6 hours, and 24 hours approximately. Infrared images were captured in still image mode using a thermal imaging camera (FLIR ONE, FLIR Systems, Inc., Wilsonville, Oregon, USA) attached to an Android-based smartphone (installed with FLIR ONE App) under ambient conditions at the bedside. A black cardboard was placed under the limbs to block thermal emissions from bedding clothes, if any (Fig 2A). Thermal range of FLIR ONE device is -20°C to 120°C with a resolution of 0.1°C and an accuracy of ±3°C. Thermal images have a spatial resolution of 160×120 pixels. The smartphone-connected FLIR ONE device has been previously validated against a hand-held high-end infrared thermography device (Thermo Tracer TH700N, Nippon Avionics Co., Ltd., Tokyo, Japan) for the assessment of pressure sores and diabetic foot [7].

Image processing and interpretation was done by a reference rater (TK) blinded to clinical data, in a single batch after all patients had been enrolled. Thermal images were processed on a desktop personal computer using FLIR Tools software (Version 5.13.17214.2001). We used the rainbow palette to false color code the thermal differences, wherein colors toward the white—red end indicate hottest regions and the blue—black end indicate coldest regions in the field of

**Table 1. Adjudication of envenomation status.** WBCT20 = 20-minute whole-blood clotting test; INR = international normalized ratio.

| Local swelling | WBCT20 | Prothrombin time | Envenomation status |
|---|---|---|---|
| Present | Did not clot | INR >1.40 | Local + systemic envenomation |
| Present | Clotted | INR ≤1.40 | Isolated local envenomation |
| Absent | Clotted | INR ≤1.40 | No envenomation (dry bite or non-venomous snakebite) |
| Absent | Did not clot | INR ≤1.40 | No envenomation (most probably false-positive WBCT20) |
| Absent | Clotted | INR >1.40 | No envenomation (?excess citrate;? subclinical envenomation) |
| Presence of neuroparalytic manifestations, acute kidney injury, microangiopathic hemolysis or capillary leak syndrome | | | Systemic envenomation |

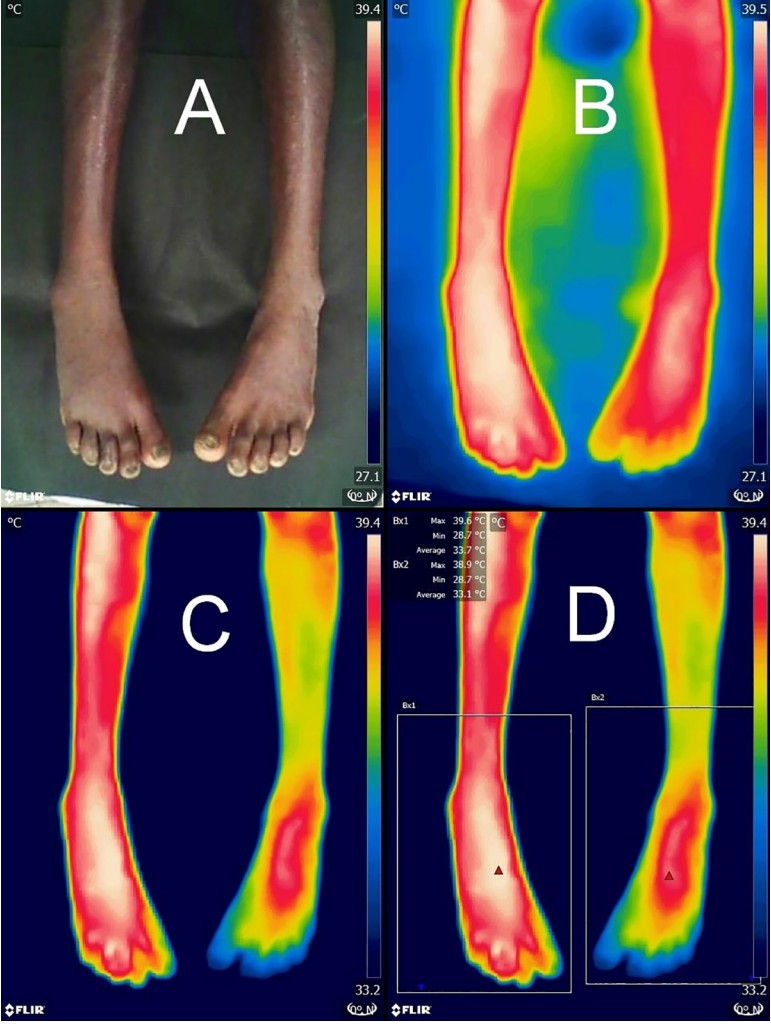

**Fig 2. Infrared thermal imaging.** Visible light image showing minimal swelling of the right foot in a patient bitten by a Russell's viper on the right third toe (Panel A). Corresponding unprocessed thermal image is shown in Panel B. Images acquired about 22 hours following the snakebite are presented here to demonstrate typical changes. Image processing was done by adjusting the lower limit of temperature range to make thermal difference between the limbs readily apparent (Panel C). Measurement of ΔTmax using box measurement tool is depicted in Panel D.

view. We manually adjusted the lower limit of temperature level so as to eliminate the thermal glow around body part being imaged (Fig 2B and 2C). This makes it easier to analyze temperature anomaly. We interpreted the images by comparing thermal pattern of the bitten body part with its contralateral counterpart simultaneously imaged under same operating conditions, known as comparative qualitative infrared thermography. This avoids the need for adjustments to compensate for environmental conditions and surface emissivity [8]. In addition, we used the box measurement tool to find out the difference in maximal temperature between the two limbs ($\Delta T_{max}$) (Fig 2D).

## Assessment of interrater and intrarater reliability

Just before reliability assessment, novice raters were briefed by the reference rater (TK) how to interpret whether a hot spot is present or not. Two representative images were used for this training. Subsequently, processed infrared thermal images acquired at enrolment from all 89

patients were independently evaluated by three novice raters. Site of bite was marked on these images; raters did not have access to any other information. Each novice rater scored the images for presence of hot spot near the bite site as present, absent, or doubtful. We estimated interrater reliability by individually comparing the assessments of novice raters with that of reference rater. To evaluate intrarater reliability, the reference rater classified the images once again after a gap of 4 weeks blinded to initial assessment.

## Sample size calculation

Since this was an exploratory study, we did not perform a formal sample size calculation. Moreover, no prior information was available to inform sample size estimation.

## Statistical analysis

We used a statistical software package (Stata/IC 12.1 for Windows, StataCorp LP, College Station, Texas, USA) for analysing data. We summarized normally distributed continuous variables as mean ± SD and continuous variables with a skewed distribution as median (IQR). We summarized categorical variables as frequency with proportion (n [%]). We compared the mean time to enrolment between two groups using independent samples *t* test. We applied Wilcoxon rank-sum test to compare the median $\Delta T_{max}$ between those with and without hot spots. We constructed 2×2 tables and calculated sensitivity, specificity, predictive values and likelihood ratios along with their 95% confidence intervals (95% CI) using an online calculator (https://www.medcalc.org/calc/diagnostic_test.php). In the main analysis, we excluded patients with unclear envenomation status. To assess robustness of diagnostic accuracy estimates, we also conducted a sensitivity analysis in which all patients with unclear envenomation status were assumed to be envenomed (worst-case scenario). In response to peer reviewers' comments, we also analyzed whether the diagnostic accuracy differed between patients in whom the snake species could be identified and the remainder. We estimated Cohen's kappa statistic, a chance-corrected measure of agreement, to evaluate interrater and intrarater reliability.

## Results

Over a period of 13 months, we enrolled 92 patients presenting to the emergency with a history of snakebite in the past 24 hours. Of them, we excluded three patients for the following reasons—in one patient infrared imaging could not be done; in another patient, the consent process was not completed; and the third patient had pre-existing skin lesions over both legs. Mean age of the included patients was 41±14 years; 60 (67%) of them were male. Median time from bite to enrolment was 11 (6.5–15) hours; 21 (24%) patients were enrolled within 6 hours of bite.

Offending snake could be identified in 44 (49%) patients. In 15 of the 44 patients, the snake was identified by examining the dead snake specimen/photograph, and in the remaining 29 patients, the snake was identified using template picture. Twelve patients were bitten by cobra (*Naja naja*), 10 patients were bitten by Russell's viper (*Daboia russelli*), and 4 patients each were bitten by saw-scaled viper (*Echis carinatus*) and krait (*Bungarus caeruleus*). In 14 patients, the offending snake was identified as non-venomous. All bites, except one, were sustained while the victims were walking through or at work in agricultural fields or near their homes located in rural areas. One patient was bitten on the forehead by a Russell's viper (*Daboia russelli*) while sleeping on the floor. Most bites (61 [69%]) were sustained on the lower limbs; 27 (30%) patients were bitten on the upper limbs, and 1 patient was bitten on the forehead. One patient was bitten on both lower limbs. Of the 89 patients, fang marks could not be made out

on examination in 34 (38%) patients. A single fang mark was evident in 28 (31%) patients; 25 (28%) had two fang marks, and 2 patients had >2 fang marks.

Of the 89 included patients, 41 patients had no local swelling at presentation. In them, the WBCT20 at presentation was abnormal in 16 patients. Of the 25 patients with a normal WBCT20, the INR was ≤1.40 in all 10 patients tested. Similarly, of the 16 patients without swelling but with abnormal WBCT20 at presentation, INR was found to be ≤1.40 in all 10 patients tested. Thus, all snakebite patients without evident swelling at the bite site had a normal INR (when tested) irrespective of the WBCT20 results (Fig 2). We classified 35 patients without evident swelling following snakebites as "No envenomation" group. This group included 25 patients with a normal WBCT20 and 10 patients with an abnormal WBCT20 but a normal INR. Remaining six patients with an abnormal WBCT20 in whom prothrombin time was not tested were classified as "Unclear envenomation status". Antivenom treatment was started in these patients before INR could be tested to rule out false-positive WBCT20. In all, 48 patients had local swelling. Among them, 40 patients had an abnormal WBCT20; however, INR was normal in 3 of the 18 patients tested. Eight patients with swelling had a normal WBCT20 at admission. We classified all the 48 patients with local swelling with/without systemic envenomation as "Envenomation present" (Fig 3). Information on pain in the bitten body part at the time of first evaluation was available in 86 patients. Of the 48 patients with envenomation, 13 had severe unbearable pain in the bitten body part, 32 had moderate but bearable pain, and 3 had mild pain. Of the 33 without envenomation, 18 did not complain of pain, 12 had mild pain, 2 had moderate but bearable pain, and 1 patient had severe unbearable pain. Of those with unclear envenomation status, 3 did not have pain, and 2 had mild pain.

Of the 48 patients with envenomation, coagulopathy evidenced by abnormal WBCT20 was present in 40 patients, and 7 patients had bleeding manifestations. Fourteen (16%) patients developed acute kidney injury; of them, five patients needed hemodialysis. Microangiopathic hemolysis was evident in 10 patients, and 3 patients developed capillary leak syndrome. Neuroparalytic manifestations were present in five patients.

At the time of presentation, area of increased temperature around the site of bite (Hot spot) was appreciable on infrared imaging in 45 of the 48 patients in the envenomed group (S1 Fig). On the other hand, hot spot was evident in only 6 of the 35 patients without envenomation (S2 Fig; Table 2). Of the three patients with envenomation but without a hot spot on infrared

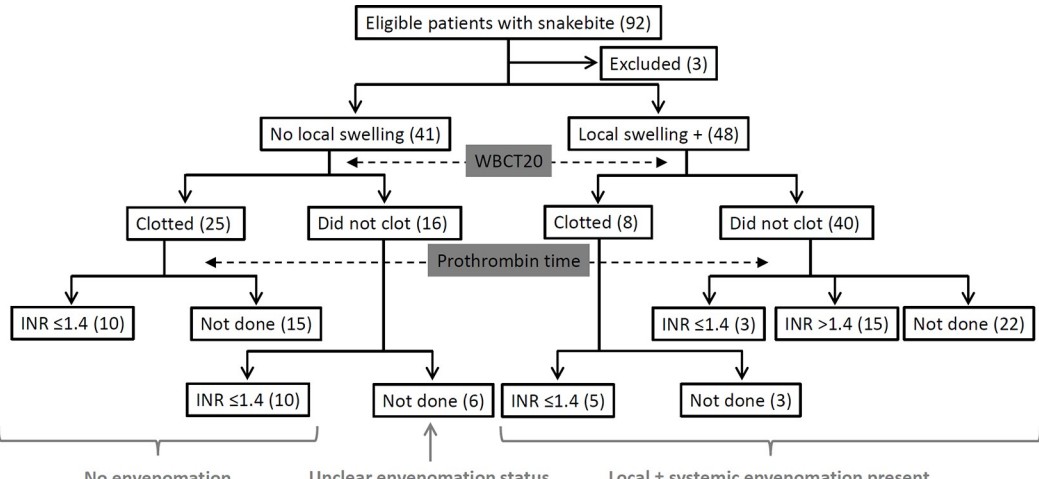

**Fig 3. Flowchart depicting classification of study patients based on local swelling and coagulopathy.**

**Table 2. Findings on infrared thermal imaging at enrolment presented by envenomation status.** * = These two findings were not mutually exclusive.

| Infrared imaging | Clinical envenomation | | |
|---|---|---|---|
| | Yes (n = 48) | Unclear (n = 6) | No (n = 35) |
| Hot spot present | 45 | 2 | 6 |
| Hot spot absent* | 3 | 4 | 29 |
| Cold spot present* | 3 | 1 | 10 |

imaging at baseline, the local swelling was limited and prothrombin time was prolonged in only one of the three patients. In that patient, a hot spot became evident 6 hours later on infrared imaging (S1 Fig; #5). Of the six patients with unclear envenomation status, hot spot was evident in two patients (S3 Fig; #29 and #34). Two of the remaining four patients developed hot spots on subsequent imaging (S3 Fig; #17 and #25; follow-up images not shown). Overall, 10 of 53 patients with hot spots on baseline imaging were evaluated within 6 hours after snakebite. Among them, hot spots were evident as early as 2½ hours following snakebite in two patients. On average, patients without hot spots on baseline infrared images had presented to the hospital earlier by about 3.3 hours as compared to those with changes on infrared imaging (36 vs 53 patients; 9.2±4.4 vs 12.5±6.4 hours; $P$ = 0.005). However, as described earlier, only 3 of the 36 patients without hot spots at baseline evaluation developed changes on infrared imaging in the next 24 hours of follow-up.

Presence of hot spot on baseline infrared images had a sensitivity of 94% and a specificity of 83% to differentiate envenomed patients from those without evidence of envenomation. Other measures of diagnostic accuracy for this comparison are presented in Table 3. After assuming the worst-case scenario, presence of hot spot on infrared imaging still had a sensitivity of 87% to identify envenomed patients; specificity remained the same (Table 3).

Relationship between hotspot on infrared thermal imaging and envenomation was equally evident in subgroups based on whether the offending snake species was identified or not and by different snake species (Table 4).

$\Delta T_{max}$ between limbs was significantly higher in those with hot spots on infrared imaging (1.10 [0.80–2.0]˚C vs -0.40 [-0.60–0.10]˚C; Fig 3; $P$ < 0.001). Except for five patients, $\Delta T_{max}$ between limbs did not exceed 3˚C in those with hot spots (Fig 4). In at least 14 of the 89 patients, we observed a localized decrease in temperature corresponding to the bite site (S1 Fig; #52) or a generalized decrease in limb temperature on the bitten side (S2 Fig; #51). Interestingly, all three envenomed patients who did not have a hot spot at baseline instead had such paradoxical thermal changes (S1 Fig).

There was more than substantial agreement between the novice raters and the reference rater (interrater reliability; Table 5), except that one of the novice raters scored a sizeable

**Table 3. Diagnostic accuracy of hot spot on infrared thermal imaging to identify snakebite envenomation.** * = Patients with unclear envenomation status were excluded from analysis; † = Patients with unclear envenomation status were assumed as envenomed.

| Measure of diagnostic accuracy | Main analysis* | Sensitivity analysis† |
|---|---|---|
| Sensitivity | 93.7% (82.8% to 98.7%) | 87.0% (75.1% to 94.6%) |
| Specificity | 82.9% (66.3% to 93.4%) | 82.9% (66.3% to 93.4%) |
| Positive predictive value | 88.2% (78.3% to 94.0%) | 88.7% (79.0% to 94.2%) |
| Negative predictive value | 90.6% (76.2% to 96.7%) | 80.6% (67.1% to 89.4%) |
| Positive likelihood ratio | 5.47 (2.63 to 11.37) | 5.08 (2.43 to 10.59) |
| Negative likelihood ratio | 0.08 (0.02 to 0.23) | 0.16 (0.08 to 0.32) |

**Table 4. Relationship between hotspot on infrared thermal imaging at enrolment and envenomation presented by snake species.**

| Snake species | Envenomation status | Hot spot on infrared thermal imaging | |
|---|---|---|---|
| | | Absent (n = 36) | Present (n = 53) |
| *Daboia russelli* (n = 10) | Yes | -- | 9/10 |
| | No | 1/10 | -- |
| *Echis carinatus* (n = 4) | Yes | 1/4 | 3/4 |
| | No | -- | -- |
| *Naja naja* (n = 12) | Yes | -- | 9/12 |
| | No | 1/12 | 2/12 |
| *Bungarus caeruleus* (n = 4) | Yes | -- | 2/4 |
| | No | 2/4 | -- |
| Non-venomous species (n = 14) | Yes | -- | 2/14 |
| | No | 12/14 | -- |
| Unidentified species (n = 45) | Yes | 2/45 | 20/45 |
| | No | 13/45 | 4/45 |
| | Unclear | 4/45 | 2/45 |

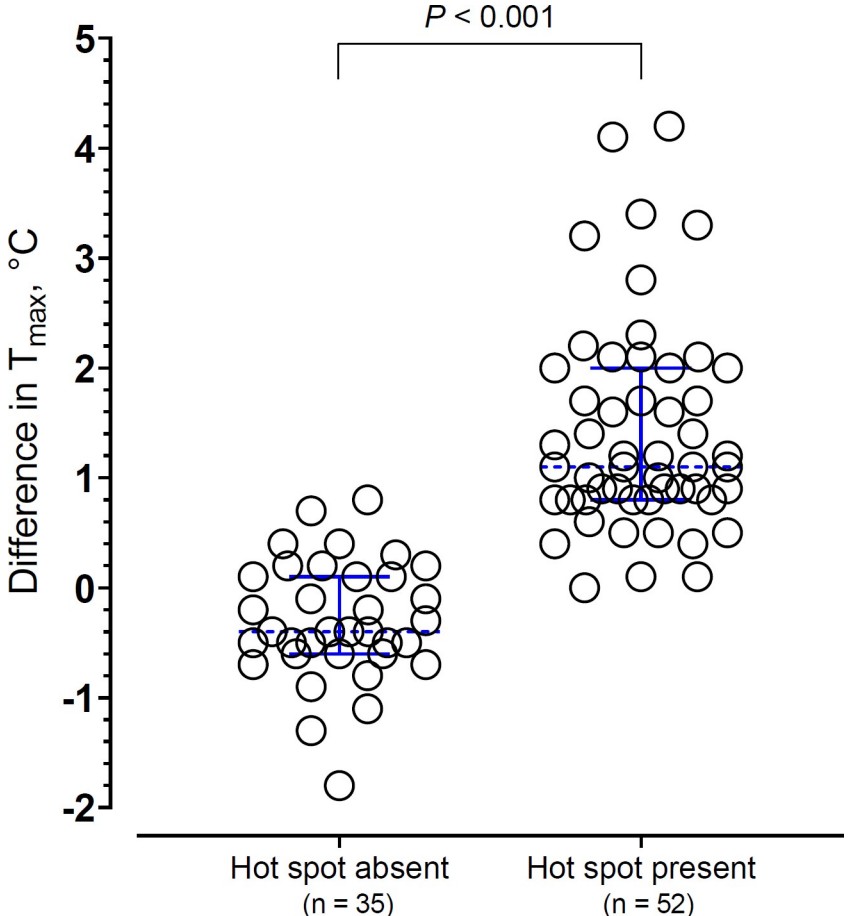

**Fig 4. Dotplot of ΔTmax presented by hot spot status.** Dotted lines across data points represent the median and error bars represent interquartile range.

**Table 5. Interrater and intrarater reliability for identification of hot spot on infrared thermal images.** * = Images scored as doubtful were excluded; † = All images included in analysis; ‡ = Only novice raters could classify an image as doubtful.

| Novice raters | Reference rater | | Kappa statistic 1* | Kappa statistic 2† |
| --- | --- | --- | --- | --- |
| | Hot spot+ | Hot spot- | | |
| *Rater 1* Hot spot + | 50 | 4 | 0.878 | 0.818 |
| Hot spot - | 1 | 31 | | |
| Doubtful‡ | 2 | 1 | -- | |
| *Rater 2* Hot spot + | 47 | 1 | 0.885 | 0.654 |
| Hot spot - | 3 | 25 | | |
| Doubtful‡ | 3 | 10 | -- | |
| *Rater 3* Hot spot + | 51 | 4 | 0.876 | 0.817 |
| Hot spot - | 1 | 30 | | |
| Doubtful‡ | 1 | 2 | -- | |
| *Reassessment by reference rater* | | | | |
| Hot spot + | 52 | 2 | 0.930 | -- |
| Hot spot - | 1 | 34 | | |

number of images (13 [15%] of 89) as doubtful. Novice raters took no more than 30 minutes to score all 89 images. Repeated assessments by the reference rater were in almost perfect agreement (intrarater reliability; Table 5).

## Discussion

We found that infrared thermal imaging done at the bedside using a smartphone-based device revealed area of increased temperature in most patients with snakebite envenomation, while only a few patients without envenomation had such changes on infrared imaging. These findings suggest that infrared thermal imaging could be useful in the evaluation of patients bitten by snakes.

Envenomations by most venomous terrestrial snakes cause swelling around the site of bite, which appears quickly within a few hours, often preceding systemic manifestations [2]. While clinical examination for size difference and local warmth are the easiest ways one could make out these changes, findings are often equivocal during the early hours following a snakebite, and the magnitude of thermal changes caused by local envenomation is considerably less compared to infective cellulitis. In a recent study [9], the mean $\Delta T_{max}$ in patients with infective cellulitis was found to be 3.7˚C (95% CI 2.7–4.8˚C), which is much higher compared to the thermal changes observed in snakebite envenomation. Hairy skin over the hands is considerably more sensitive than glabrous skin over the palms [10]. However, even trained observers have difficulty in correctly identifying temperature differences less than 3˚C by palpation [11]. We demonstrate that temperature differences following snakebite envenomation often not exceeding 3˚C could be made readily apparent by using infrared thermal imaging. Further, infrared thermal imaging had a high interrater and intrarater reliability, making it a more objective way of assessing local changes caused by snakebite envenomation.

Published studies on the use of infrared thermal imaging in patients bitten by venomous animals and insects are very few. For the first time, Medeiros et al. [5] described the use of infrared imaging to identify thermal changes in a small case series of patients with snakebites (n = 3), spider bites (n = 3) and scorpion stings (n = 2). In this series, a patient with dry bite by lance-headed pit viper (*Bothrops jararaca*) did not show thermal changes on infrared images, whereas the other two patients with local envenomation caused by rattlesnake (*Crotalus durissus terrificus*) and lance-headed pit viper (*Bothrops moojeni*) had thermal changes [5].

Subsequently, infrared thermal imaging findings in a patient with coral snake (*Micrurus frontalis*) envenomation have been described [12]. We extend these anecdotal observations by demonstrating that presence of thermal changes detected using infrared imaging could be used to differentiate venomous snakebites from non-venomous and dry bites.

We found that infrared thermal imaging had a high sensitivity and modest specificity to differentiate between dry/non-venomous bites and venomous snakebites. This means that envenomation is unlikely in the absence of thermal changes. On the other hand, most patients with thermal changes developed features of local envenomation with or without systemic envenomation. Further, the present study has brought out certain interesting thermographic changes characterized by comparatively lower temperature on the bitten limb, which have not been described before. Mechanism and clinical significance of such paradoxical changes are, however, unclear at present. Of note, such changes were observed in all the three patients with envenomation who did not have hot spots on baseline images.

We observed that thermal changes were apparent in some patients without envenomation. True meaning of these thermal changes in the absence of envenomation is unclear. It is possible that these patients were inoculated with a low quantum of venom that was not sufficient enough to result in clinical envenomation. We suggest that infrared imaging could help in early identification of dry/non-venomous bites among those without evidence of envenomation at presentation. Presently, experts recommend that such patients should be observed for 24 hours at least [13,14]. Likewise, infrared imaging could be useful when the WBCT20 is abnormal in the absence of local swelling. This discordant situation is not uncommon. Despite the simplicity of WBCT20, it is well recognized that false-positive WBCT20 is a common clinical phenomenon. Dsilva et al. observed that 24% of abnormal WBCT20 results were false-positive [4]. Rathnayake et al. found that 13 of 908 snakebite patients with INR <1.4 had a false-positive WBCT20 [3]. In such situations, estimation of prothrombin time would be required. However, laboratory facilities to estimate prothrombin time may not be available in all clinical settings, and even if available results of prothrombin time might not be available timely to inform decision making. Present findings suggest that envenomation is unlikely in such patients if thermal changes are absent. On the other hand, infrared imaging would add little diagnostic value in patients with clear evidence of local and/or systemic envenomation.

One important question is how early these thermal changes appear. While the present study cannot answer this question directly, we observed thermal changes as early as 2½ hours after snakebites. In a case report, Medeiros et al. have noted thermal changes 10 minutes after a coral snake bite [12]. Another important question is the cost and feasibility of infrared imaging in resource-limited settings. The imaging device used in our study costs a few hundred US dollars. However, this device is reusable and does not require consumables. Therefore, actual cost per patient would be very much less (about USD 1 per patient). In comparison, an average dose of 20 vials of antivenom costs about USD 70 per patient in our setting. Further, a smartphone App is now available to process images, obviating the need for a computer and reliable electrical power supply. Thus, it is feasible to acquire, process and interpret infrared images at the bedside within 20 minutes, the time taken for WBCT20 testing.

The present study has some strengths—i) We enrolled a large number of patients prospectively, including patients with varying degree of envenomation and those with unclear envenomation status, thereby avoiding spectrum bias; ii) We performed infrared imaging at the bedside under ambient conditions to enhance generalizability; and iii) We formally evaluated interrater and intrarater reliability of infrared image interpretation. There are some possible limitations to our study. First, although we included all snakebites irrespective of the species, krait envenomation was not sufficiently represented in our study sample. Therefore, we do not know whether patients with krait envenomation, who typically have little local changes, would

also have demonstrable changes on thermal imaging. Second, some NFFC snakebites might result in local changes that could be misdiagnosed as isolated local envenomation in the absence of snake identification. In fact, we observed that two patients bitten by non-venomous snakes had local envenomation which was picked up on infrared imaging as well. While it might seem that these two instances are false-positives, it should be noted that at least five species of NFFC snakes are recognized as potentially lethal, and specific antivenom treatment is recommended for envenomation caused by three NFFC species [15]. Given this, it may not be appropriate to classify such instances as false-positives. Rather, all snakebite victims with signs of local envenomation would need careful monitoring for systemic envenomation, irrespective of the identity of the offending snake.

Third, we could not objectively verify envenomation by using a venom assay due to non-availability. Fourth, presence of thermal changes in the bitten limb does not give any information about systemic envenomation. However, for most of the venomous terrestrial snakes in our locale except the kraits, systemic manifestations are unlikely in the absence of local envenomation. Although systemic envenomation without evidence of local envenomation has been reported in Russell's viper bites [13,16], recent studies suggest that such discordant instances could be due to false-positive WBCT20 [4]. Moreover, the envisaged clinical application of infrared imaging would be to rule out, rather than to rule in, envenomation in patients with snakebites. Future studies are required to evaluate whether infrared thermal imaging could be useful in primary care settings to risk stratify patients presenting very early following snakebites and in the evaluation of bites by a broader range of snake species.

## Conclusions

We found that snakebite envenomation was accompanied by temperature changes over the bitten body part which could be readily and reproducibly identified using infrared thermal imaging. We also found that envenomation was unlikely in the absence of thermal changes. Point-of-care infrared thermal imaging could be a useful adjunct to the standard evaluation of patients bitten by snakes, particularly in early recognition of non-venomous and dry bites.

## Supporting information

**S1 Fig. Infrared thermal images acquired at enrolment in patients with envenomation.** Approximate site of snakebite is indicated using white circles. Study enrolment numbers are presented alongside images. Patient #75 was bitten over both legs.
(PDF)

**S2 Fig. Infrared thermal images acquired at enrolment in patients without envenomation.** Approximate site of snakebite is indicated using white circles. Study enrolment numbers are presented alongside images.
(PDF)

**S3 Fig. Infrared thermal images acquired at enrolment in patients with unclear envenomation status.** Approximate site of snakebite is indicated using white circles. Study enrolment numbers are presented alongside images.
(PDF)

## Acknowledgments

Authors thank all patients who participated in this study.

## Author Contributions

**Conceptualization:** Tamilarasu Kadhiravan.

**Data curation:** Paramasivam Sabitha, Tamilarasu Kadhiravan.

**Formal analysis:** Surendran Deepanjali, Bettadpura Shamanna Suryanarayana, Tamilarasu Kadhiravan.

**Investigation:** Paramasivam Sabitha.

**Methodology:** Paramasivam Sabitha, Chanaveerappa Bammigatti, Tamilarasu Kadhiravan.

**Project administration:** Tamilarasu Kadhiravan.

**Resources:** Tamilarasu Kadhiravan.

**Supervision:** Chanaveerappa Bammigatti, Tamilarasu Kadhiravan.

**Validation:** Chanaveerappa Bammigatti, Surendran Deepanjali, Bettadpura Shamanna Suryanarayana.

**Writing – original draft:** Paramasivam Sabitha, Tamilarasu Kadhiravan.

**Writing – review & editing:** Chanaveerappa Bammigatti, Surendran Deepanjali, Bettadpura Shamanna Suryanarayana.

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
