## [Decision Letter · Decision Letter 0]

10 Nov 2020

Dear Dr Kadhiravan,

Thank you very much for submitting your manuscript "Smartphone-based infrared thermal imaging for differentiating venomous snakebites from non-venomous and dry bites" for consideration at PLOS Neglected Tropical Diseases. As with all papers reviewed by the journal, your manuscript was reviewed by members of the editorial board and by several independent reviewers. In light of the reviews (below this email), we would like to invite the resubmission of a significantly-revised version that takes into account the reviewers' comments. 

We cannot make any decision about publication until we have seen the revised manuscript and your response to the reviewers' comments. Your revised manuscript is also likely to be sent to reviewers for further evaluation.

Sincerely,

Abdulrazaq G. Habib

Guest Editor

José Gutiérrez

Deputy Editor

Reviewer's Responses to Questions

**Key Review Criteria Required for Acceptance?**

**Methods**

-Are the objectives of the study clearly articulated with a clear testable hypothesis stated?

-Is the study design appropriate to address the stated objectives?

-Is the population clearly described and appropriate for the hypothesis being tested?

-Is the sample size sufficient to ensure adequate power to address the hypothesis being tested?

-Were correct statistical analysis used to support conclusions?

-Are there concerns about ethical or regulatory requirements being met?

Reviewer #1: Are the objectives of the study clearly articulated with a clear testable hypothesis stated?

Yes

-Is the study design appropriate to address the stated objectives?

Not completely

-Is the population clearly described and appropriate for the hypothesis being tested?

Yes

-Is the sample size sufficient to ensure adequate power to address the hypothesis being tested?

Yes

-Were correct statistical analysis used to support conclusions?

Yes

-Are there concerns about ethical or regulatory requirements being met?

No

Reviewer #2: In my opinion, the study was appropriate and correctly performed. 

The references are compatible with the text.

Reviewer #3: Objectives are clear and hypothesis is testable. 

Study design is reasonable but I question the Adjudication of Envenomation methods of determining envenomation. This algorithm is an essential aspect to the study but needs referencing, especially for venomous species found in this region of southern India, if this available. Can also extrapolate other similar studies but the reference by Gutierrez et al. is a review article that references a couple other studies in regards to diagnosis. Population sampled needs to be explained better. When they say "definitive" snake bite what does that mean. Did the individual need to see the snake bite them? The reason this is important because only 49% were able to properly able to identify a snake. 

Methods needs to utilize Genus and Species or Genus, for venomous snakes in this region not just common name. I would also make a statement of how many venomous species are present in this region as well. Statistical analysis seems reasonable for this study. No ethical concerns are perceived and each participant underwent informed consenting.

**Results**

-Does the analysis presented match the analysis plan?

-Are the results clearly and completely presented?

-Are the figures (Tables, Images) of sufficient quality for clarity?

Reviewer #1: -Does the analysis presented match the analysis plan?

Yes

-Are the results clearly and completely presented?

Yes

-Are the figures (Tables, Images) of sufficient quality for clarity?

Yes

Reviewer #2: Yes

Reviewer #3: Analysis matches the plan. Majority of results are clear. One important aspect of this study is correlation with known venomous and non-venomous snake bite encounter, and results of adjudication of envenomation algorithm. It looks like only 49% of the cohort was able to identify the snake. This was touched in the discussion as a limitation but the data needs to be more clearly reported. It is true that some individuals bitten cannot recall the snake or recognize the offending snake, or the encounter was in the setting of not seeing the snake bite, but it is important to discuss among those who were not able to identify the offending snake, who developed swelling, coagulopathy...as we are assessing all potential scenarios. With that being said, Table 1. needs referencing and validation as a tool for ruling in or ruling out envenomation with vipers and cobras/kraits. Population bitten needs to be explained in results. For example, occupation, encounter the led to envenomation (step on the snake, handling a snake...) as this supports that among the group who could not identify the snake, that they were actually bitten. I also recommend correlating the data to the known venomous species that were identified. In my opinion this is very useful for your manuscript and would support the Adjudication of Envenomation algorithm with more rigor. Also, if pain scale was collected this would be useful as well. Most envenomation's are painful to some extent and typically worsens during the first 24 hours compared to non-venomous snake bite.

**Conclusions**

-Are the conclusions supported by the data presented?

-Are the limitations of analysis clearly described?

-Do the authors discuss how these data can be helpful to advance our understanding of the topic under study?

-Is public health relevance addressed?

Reviewer #1: Please see the review below

Reviewer #2: Yes

Reviewer #3: I agree that this technology could be useful in low-resource settings such as in this situation. I would mention that in the conclusion, as the need to decide when to utilize antivenom is not always evident but this tool could be a way to help make that decision. Another thing to mention would be why the adjudication of envenomation was created for this study, as you could have excluded all those who could not identify the snake from the analysis. Also, other laboratory findings, such as inflammatory and coagulation markers are used in monitoring those with snakebite envenomation, and this needs to be explained in the discussion. If not available then should be mentioned. Those three markers are useful but I am not aware of a study only using these perimeters.

**Editorial and Data Presentation Modifications?**

Reviewer #1: (No Response)

Reviewer #2: Detailed comments:

Line 24 and 113-114. 6 and 24 hours after the first assessment and not from the bite - because you can understand ambivalently? Better to add "later" as the explanatory word.

Line 126. It's worth adding the time from the bite to taking the photo.

Line. 201-202. Please add an explanation why 6 patients did not have INR performed.

Reviewer #3: Some grammatical and spelling errors exist.

**Summary and General Comments**

Reviewer #1: In this manuscript by Sabita et al., thermal imaging of snakebite victims is being proposed as a strategy to distinguish venomous snakebites from dry bites or bites from clinically unimportant snakes. While this strategy, which has been proposed before, can be useful for diagnosing snakebites, its application in the context of snakebite management in India is questionable. Among other reasons, the time taken by most snakebite victims to reach the hospital is such large that they are likely to exhibit one or more adverse effects of envenomation. Moreover, false positives associated with thermal imaging is not significantly smaller than those exhibited by PT/INR and WCT20. Hence, the application of this very expensive technology in diagnosing venomous bites from dry bites or bites from clinically unimportant snakes is debatable.

Major comments

• Throughout the manuscript, the authors refer to snakes and snakebites as “poisonous”. This needs to be corrected as this is scientifically inaccurate. Most snakes are venomous, and not poisonous. Similarly, non-venomous could be replaced with ‘not medically/clinically important’.

• Venom is a shared trait amongst snakes and probably evolved in the common ancestor of the clade Serpentes (Fry et al. 2013). It is, therefore, not surprising that venom protein homologues have also been detected in species that are medically unimportant to humans (e.g., most colubrid snakes; see Lumsden et al. 2005) and in snakes that are not known to rely on venom for prey capture (e.g., Henophidia; Fry et al. 2013). The presence of SVMPs, 3FTxs, PLA2s and other major venom proteins in colubrids is a well-known fact. Given this, it is likely that colubrid snakebite victims too can exhibit these local symptoms, including swelling and localized temperature changes. This is particularly true for bites from vine snakes, trinkets, mud snakes, and probably even keelbacks. If recruiting bite victims of these snakes is not possible, then the authors should at least clarify this aspect in their manuscript and highlight situations that could result in false positives. 

• As the authors report, the majority of snakebite victims reach the hospital several hours post bite (only 24% reached the hospital within 6 hours of bite in this study). In case of serious envenomations, bite victims develop typical symptoms of envenomation in a lot less time. Considering this, how will this technique be useful?

• The title is slightly misleading as this is not really a ‘cell phone based’ technology (most smartphones don’t have thermal cameras). It relies on a very expensive external device. Data processing is done outside the cell phone too. The authors should clarify this in the paper and the title should be revised accordingly.

• Though the percentage of false positives in the case of WBCT20 (24%) is higher than thermal imaging [17% (6/35 cases)], what are the chances that both WBCT20 and PT/INR would yield false positives? The discussion provided for the advantage of infrared imaging over conventional PT/ INR based detection is not convincing, especially after considering the cost of the thermal imaging system.

Minor comments

• Lines 47 to 39: If no abnormalities are found in the blood test, then is it appropriate to consider bites as ‘non-venomous’ or dry bites? What about neurotoxic snakebites? This needs to be clarified.

• Line 53: false positives reinstate why it is important to recruit bite victims of medically unimportant snakes in this study. See my major comment above.

• Line 66: It is a misconception that antivenoms are unavailable in India. They are manufactured in sufficiently large quantities. Lack of antivenoms in hospitals is often due to administrative reasons. This sentence, therefore, needs to be modified appropriately. In fact, the authors should point out the drawbacks associated with unnecessary administration of antivenoms.

References

Lumsden et al. 2005. Pharmacological characterisation of a neurotoxin from the venom of Boiga dendrophila (mangrove catsnake)

Fry et al. 2013. Squeezers and leaf-cutters: differential diversification and degeneration of the venom system in toxicoferan reptiles

Reviewer #2: As in the letter

Reviewer #3: Overall the study is interesting and reasonable. In low-resource settings, this technology may be useful and this should be one of the talking points. Data presented also seems reasonable but I suggest focusing on those who we know had true envenomation as they could identify the culprit snake. You could also compare known non-venomous encounter from the study. The reason why this is so important is people will commonly state they were bitten by something and blame it on a snake, typically a venomous snake. The reason why I am harping on the Adjudication of Envenomation algorithm is I have seen plenty of people who present atypically when bitten by known venomous snake, you have to be careful as a clinician. If a patient was known to be bitten by a venomous snake, regardless of whether it is "dry" or not, we typically offer antivenom while monitoring. In low-resource settings, and even in my own practice, we have seen shortage of antivenom and/or having to decide on when to use as the story was not that reliable. I could see infrared technology if validated with more rigor being a possible deciding factor as opposed to watchful clinical and laboratory monitoring.

PLOS authors have the option to publish the peer review history of their article (what does this mean?). If published, this will include your full peer review and any attached files.

Reviewer #1: No

Reviewer #2: Yes: Piotr Borkowski

Reviewer #3: No
---

## [Decision Letter · Decision Letter 1]

1 Feb 2021

Dear Dr Kadhiravan,

We are pleased to inform you that your manuscript 'Point-of-care infrared thermal imaging for differentiating venomous snakebites from non-venomous and dry bites' has been provisionally accepted for publication in PLOS Neglected Tropical Diseases.

Best regards,

Abdulrazaq G. Habib

Guest Editor

José Gutiérrez

Deputy Editor

Reviewer's Responses to Questions

**Key Review Criteria Required for Acceptance?**

**Methods**

-Are the objectives of the study clearly articulated with a clear testable hypothesis stated?

-Is the study design appropriate to address the stated objectives?

-Is the population clearly described and appropriate for the hypothesis being tested?

-Is the sample size sufficient to ensure adequate power to address the hypothesis being tested?

-Were correct statistical analysis used to support conclusions?

-Are there concerns about ethical or regulatory requirements being met?

Reviewer #1: (No Response)

Reviewer #3: Revision has addressed previous critique. Objectives are met.

**Results**

-Does the analysis presented match the analysis plan?

-Are the results clearly and completely presented?

-Are the figures (Tables, Images) of sufficient quality for clarity?

Reviewer #1: (No Response)

Reviewer #3: Analysis has bene revised based of reviewers recommendations and explanations are sufficient.

**Conclusions**

-Are the conclusions supported by the data presented?

-Are the limitations of analysis clearly described?

-Do the authors discuss how these data can be helpful to advance our understanding of the topic under study?

-Is public health relevance addressed?

Reviewer #1: (No Response)

Reviewer #3: Revision provided addresses earlier concerns about the conclusion of the findings. In my opinion, the conclusion made is supported by the data provided.

**Editorial and Data Presentation Modifications?**

Reviewer #1: (No Response)

Reviewer #3: (No Response)

**Summary and General Comments**

Reviewer #1: I find all responses to my queries satisfactory. But note a couple of points below.

To my comment on false positives, the authors have responded that NFF colubrids are also capable of inflicting clinically severe envenoming. I am confident that the authors know that such snakes are only restricted to NE India and that the antivenoms against such snakes are largely unavailable. Nonetheless, my comment was only w.r.t NFF colubrids presenting false positives in the majority of the country, where such unique snakes are absent.

References: No, I did not suggest citing any reference. I had provided complete references of the papers that I had mentioned in my comments for the authors.

Reviewer #3: Overall the revision was well thought and sufficient explanations were made for each comment left by the reviewers. Authors have revised the manuscript in a way that allows the collected data to be presented appropriately.

PLOS authors have the option to publish the peer review history of their article (what does this mean?). If published, this will include your full peer review and any attached files.

Reviewer #1: No

Reviewer #3: No

---

## [Editor Report · Acceptance letter]

13 Feb 2021

Dear Dr. Kadhiravan,

We are delighted to inform you that your manuscript, "Point-of-care infrared thermal imaging for differentiating venomous snakebites from non-venomous and dry bites," has been formally accepted for publication in PLOS Neglected Tropical Diseases.

Best regards,

Shaden Kamhawi

co-Editor-in-Chief

Paul Brindley

co-Editor-in-Chief
